# Reactive nitrogen restructures and weakens microbial controls of soil $N_2O$ emissions

Christopher M. Jones[1,2], Martina Putz[1,2], Maren Tiemann[1] & Sara Hallin [1✉]

The global surplus of reactive nitrogen ($N_r$) in agricultural soils is accelerating nitrous oxide ($N_2O$) emission rates, and may also strongly influence the microbial controls of this greenhouse gas resulting in positive feedbacks that further exacerbate $N_2O$ emissions. Yet, the link between legacy effects of $N_r$ on microbial communities and altered regulation of $N_2O$ emissions is unclear. By examining soils with legacies of $N_r$-addition from 14 field experiments with different edaphic backgrounds, we show that increased potential $N_2O$ production is associated with specific phylogenetic shifts in communities of frequently occurring soil microbes. Inputs of $N_r$ increased the complexity of microbial co-association networks, and altered the relative importance of biotic and abiotic predictors of potential $N_2O$ emissions. Our results provide a link between the microbial legacy of $N_r$ addition and increased $N_2O$ emissions by demonstrating that biological controls of $N_2O$ emissions were more important in unfertilized soils and that these controls are weakened by increasing resource levels in soil.

[1] Swedish University of Agricultural Sciences, Department of Forest Mycology and Plant Pathology, Box 7026, 750 07 Uppsala, Sweden. [2] These authors contributed equally: Christopher M. Jones, Martina Putz. ✉email: Sara.Hallin@slu.se

Surplus nitrogen (N) is one of the major threats to ecosystem integrity[1]. Agriculture is the main source of global N pollution[2] as well as increased atmospheric concentrations of the greenhouse gas nitrous oxide ($N_2O$), mainly through application of N fertilizers[3]. Yearly increases in global $N_2O$ emission rates are accelerating, which is consistent with the growing surplus of reactive nitrogen ($N_r$) in agricultural soils[4]. Legacy effects of elevated levels of $N_r$ include profound shifts in the structure of soil microbial communities[5,6], which are the primary drivers of $N_2O$ emissions through transformations of inorganic nitrogen species[7]. However, whether the effect of $N_r$ legacy on $N_2O$ emissions is mainly driven by abiotic factors associated with shifts in soil properties, or by changes in biotic factors, i.e. microbial community members and functional groups that directly or indirectly exert microbial controls of $N_2O$ emissions, remains uncertain.

Microbial communities that perform denitrification are the predominant source of $N_2O$ in arable soils[8–10]. Denitrifying communities remove up to 56% of newly fixed $N_r$ annually at the global scale and ~8% of the total denitrification flux results in $N_2O$[8]. Denitrification is the stepwise reduction of $NO_3^-$ to $N_2$ and is best described as a modular pathway, with different microorganisms being capable of performing all or only a subset of the reductive steps in the pathway[11]. Comparison of genomes and isolates reveal that only a minority of denitrifiers completely reduce $NO_3^-$ to $N_2$, and therefore produce $N_2O$ as a terminal product[12–14]. At the same time, the only known sink of $N_2O$ on Earth is its reduction to $N_2$ by denitrifying or non-denitrifying microorganisms that possess the gene *nosZ*, encoding the $N_2O$ reductase[15]. Changes in the proportion of producers and consumers of $N_2O$ play a causal role in determining $N_2O$ emissions[16,17], and the capacity for a microorganism to act as a producer or consumer of $N_2O$ is not randomly distributed across taxonomic groups[12]. Thus, shifts in the composition of microbial communities or co-occurrences of organisms from different taxonomic groups may predict whether the soil is more likely to act as a source or a sink of $N_2O$ in response to chronically elevated levels of $N_r$. Previous work has shown that long-term N addition increases the abundance of different bacterial phyla or classes, and it is hypothesized that organisms within these taxonomic groups share specific life-history traits that are favoured when N availability is high[18,19]. However, work using a phylogeny-based approach demonstrated that the response to elevated $N_r$ is not consistent within broadly defined taxonomic groups, and is likely conserved only to the genus level[5]. This suggests that changes in the abundances of taxonomic groups alone are not accurate predictors of $N_2O$ emission potential. Nitrogen addition can also indirectly modify the microbial community since $N_r$ promotes primary production and increases resource levels in soils, which alter microbial co-associations[20,21]. Resource-driven shifts in co-association may arise from a combination of ecological mechanisms, such as changes in antagonistic and mutualistic interactions amongst organisms, or altered environmental constraints that define shared niche preferences across species. Shifts in co-association are best assessed by analysis of microbial networks, which allow us to observe how changes in co-associations may affect emergent properties of the community. Changes in network structure, such as the number of connections between community members, number of defined communities within a network, or restructuring of co-associations may be linked to changes in ecosystem functioning[22]. In the case of $N_2O$-related functioning in soils, changes in co-associations between organisms with incomplete denitrification pathways may be of particular importance. However, the degree to which $N_r$ addition alters microbial co-association networks, and whether such shifts also fundamentally restructure microbial controls of $N_2O$ emissions, is unknown.

We address these uncertainties by examining the impact of long-term mineral N inputs on microbial communities in arable soils in 14 different long-term (15–57 years), replicated fertilization field trials in which nitrate-based mineral N fertilizers have been added yearly to arable soils at rates of 80 to 150 kg ha$^{-1}$ (Supplementary Table 1). Using multiple experiments across different soil types allowed us to identify broadly conserved effects of long-term $N_r$ addition in arable soils. Management was similar across the experiments regarding the addition of mineral fertilizers, mouldboard ploughing and annual crop rotations, which altogether minimize context-dependent effects that complicate efforts to identify general microbial responses to elevated $N_r$. We first verified the expected fertilization effects on denitrification and $N_2O$ production rates, then focused on shifts in the phylogenetic structure and patterns of co-association in overall microbial communities, as well as the abundance of genes that indicate the capacity for production and consumption of $N_2O$ in arable soils by denitrifiers or non-denitrifying $N_2O$ reducing organisms. We hypothesize that (i) there is a generic response, irrespective of site and soil physico–chemcial properties, to long-term addition of $N_r$ showing phylogenetically conserved shifts in community structure that are linked to differences in $N_2O$ emission potential and (ii) long-term $N_r$ addition restructures microbial co-associations due to increased resource availability, thereby altering the relative importance of biotic and abiotic factors in predicting whether a soil acts as a source or sink for $N_2O$. Our results show that long-term addition of $N_r$ changes community phylogenetic composition and increases the complexity of microbial networks, which in turn alters how microbial communities regulate the production of $N_2O$. Abiotic predictors of potential $N_2O$ production in relation to total denitrification rates were more important in fertilized soils than unfertilized soils. Thus, fertilization reduces the relative importance of biotic controls of $N_2O$ emissions, suggesting that biotic controls are weakened by elevated N levels.

## Results and discussion
**Addition of $N_r$ increases denitrification product ratio, adds resources and changes genetic controls of $N_2O$ emissions**. As expected, potential denitrification and $N_2O$ production rates were significantly higher in fertilized soils across all sites (Fig. 1a; Supplementary Table 2). Although this could be a direct effect of $N_r$ input, the increased denitrification end-product ratio ($N_2O$/[$N_2 + N_2O$]) in the fertilized soils indicates a change in the controls of $N_2O$ emissions (Fig. 1a). Over time, application of certain N fertilizers can decrease soil pH[23], which is known to be a strong abiotic determinant of denitrification end-product ratios[24]. However, soil pH was not significantly altered in the fertilized soils across the sites (Supplementary Table 3), and analysis of covariance (ANCOVA) indicated that the effect of pH was not different between unfertilized and fertilized soils ($F_{1,104} = 2.79$; P = 0.097; standardized regression coefficients $\beta = -0.66$ and $\beta = -0.7$ for unfertilized and fertilized soils, respectively). Thus, pH does not explain the general increase in end-product ratio observed in the fertilized soils. Long-term addition of $N_r$ increased soil organic C and N content as well as ammonium and nitrate levels and decreased the C/N ratio (Supplementary Table 3) which could support increased $N_2O$ production[25,26]. We observed no relationship between end-product ratios and soil C/N or nitrate levels, whereas potential activities and end-product ratios were weakly correlated with soil organic C, total N and $NH_4^+$ content (Spearman's $\rho = 0.25$, $0.29$ and $0.24$, respectively; P < 0.05). The increased end-product ratio may also be driven by a functional shift in the microbial communities controlling net $N_2O$ production, as suggested by

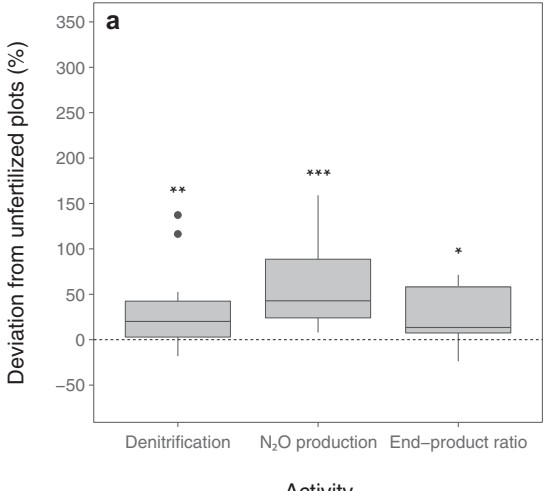 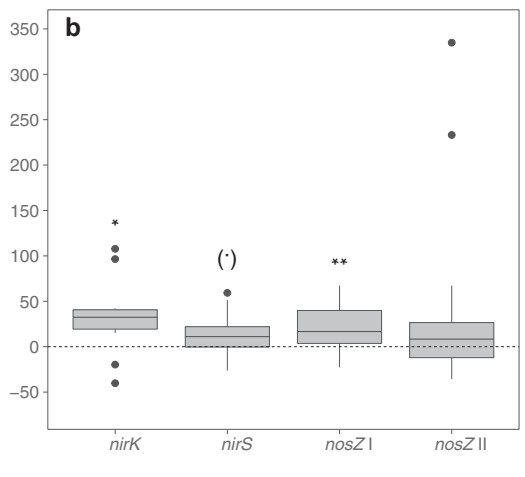

**Fig. 1 Percentage change in mean functional and genetic potentials for denitrification and N2O reduction in soils under long-term N fertilization compared to unfertilized soils across 14 different long-term experiments. a** Percent change in potential denitrification activity, potential net $N_2O$ emissions and the ratio of denitrification end products in fertilized plots compared to unfertilized control plots. **b** Percent change in functional gene relative abundances (copies per 16S rRNA gene copies) in fertilized plots compared to unfertilized control plots. Asterisks indicate significance of deviation from zero (Wilcoxon signed-rank test [$N = 14$], (.) $p < 0.1$, *$p < 0.05$, **$p < 0.01$, ***$p < 0.001$). Box limits represent the inter-quartile range (IQR) with median values represented by the centreline. Whiskers represent values ≤1.5 times the upper and lower quartiles, while points indicate values outside this range.

changes in the genetic controls of $N_2O$ emissions via denitrification. The abundance of the gene *nirK*, encoding the copper nitrite reductase in denitrifiers, increased in fertilized soils whereas no increase was observed for the *nirS* gene encoding the heme-based nitrite reductase (Fig. 1b; Supplementary Table 4). This corresponds to site-specific studies that have shown *nirK*- but not *nirS*-containing denitrifiers drive $N_2O$ production[27–29], and the previously discussed niche differentiation between denitrifiers with *nirS* vs. *nirK*[30,31]. Genome comparisons have further shown that the majority of *nirK*-type denitrifying species would produce $N_2O$ as a terminal product[12], which may in part explain the observed increase in both $N_2O$ production and ultimately end-product ratios. The increase in total abundances of only *nosZ* clade I, coding for the clade I-type $N_2O$ reductase, in fertilized plots also suggests a structural shift in functional microbial communities regulating $N_2O$ emissions, as well as niche differentiation between the major $N_2O$ reducing communities (Fig. 1b; Supplementary Table 4). Overall, the differences between fertilized and unfertilized soils show that both abiotic and biotic controls of denitrification end-product ratios were modified by the addition of $N_r$, and simple correlations cannot tease apart these effects. Furthermore, measurement of direct genetic controls may not capture the full scope of biotic controls of potential $N_2O$ emission, as changes in community composition can indirectly regulate denitrification activity.

**N-induced shifts in phylogenetic composition link to denitrification end-product ratios.** We then examined the effect of long-term addition of $N_r$ on microbial community composition using a phylogeny-aware compositional approach[32], which allowed us to identify clades driving $N_r$-induced compositional shifts, as well as accounting for the compositional nature of microbial community data. We focused our analysis on frequently occurring OTUs, as defined by species abundance distributions[33], across all soils to reduce the influence of site-specific differences in community composition. As suggested by the functional gene abundances, community composition differed between unfertilized and fertilized soils (site-constrained perMANOVA $R^2 = 0.013$, $P < 0.001$) despite strong site-specific effects (unconstrained $R^2 = 0.694$; $P < 0.001$; Supplementary

Fig. 1). Fertilization did not affect species richness or phylogenetic diversity, whereas Shannon diversity was only slightly higher (0.5%) in the fertilized soils (Supplementary Table 5). These results reflect studies showing that long-term $N_r$ addition modifies the structure of soil microbial communities, whereas effects on alpha-diversity may depend more on local conditions[19], e.g. availability of other macronutrients[34]. Closer inspection of phylogenetic changes showed a shift towards decreased abundances of Cyanobacteria, Gemmatimonadetes, Nitrospirae and Planctomycetes in response to long-term inputs of $N_r$ (Fig. 2). These changes were largely consistent across lineages within each clade, suggesting a degree of ecological coherence amongst members within each phylum in relation to soil N levels. Thereby, these shifts correspond to previous observations of taxonomic shifts in arable soils and managed grasslands[18,19,35]. By contrast, our results show that the increased abundances of Proteobacteria, Firmicutes, Actinobacteria and Bacteroidetes observed in fertilized soils varied amongst lineages within each phylum. For example, the expected overall increase in Actinobacteria[18,19,35] was driven by the increase of a few abundant lineages, whereas the majority actually decreased in the fertilized soils. Similarly, the overall decrease of Acidobacteria in fertilized soils, in agreement with other studies[5,18,19], was not consistent within this phylum as several lineages, including Thermoanaerobaculia, the Solibacter sub-lineage of the Solibacterales, and Acidobacterial Subgroups 6, 17 and 25 increased in response to long-term fertilization. These shifts are in line with reports showing these subgroups to be more abundant in soils with higher C and N availability[36,37]. We also noted that a single and frequently occurring OTU of an ammonia-oxidizing bacteria (*Nitrosospira*) increased with fertilization. Since they can produce $N_2O$, this could potentially add to elevated in situ emissions from fertilized soils, and recent work suggests that ammonia oxidation likely contributes 0.1–10% of possible maximum $N_2O$ emission rates[10]. Overall, the observed shifts underscore that traits that determine how the microbial community responds to elevated $N_r$ inputs occur at different phylogenetic scales amongst community members, and are not necessarily inferred from taxonomic affiliation. This is particularly relevant for denitrification, in which various evolutionary processes have played more or

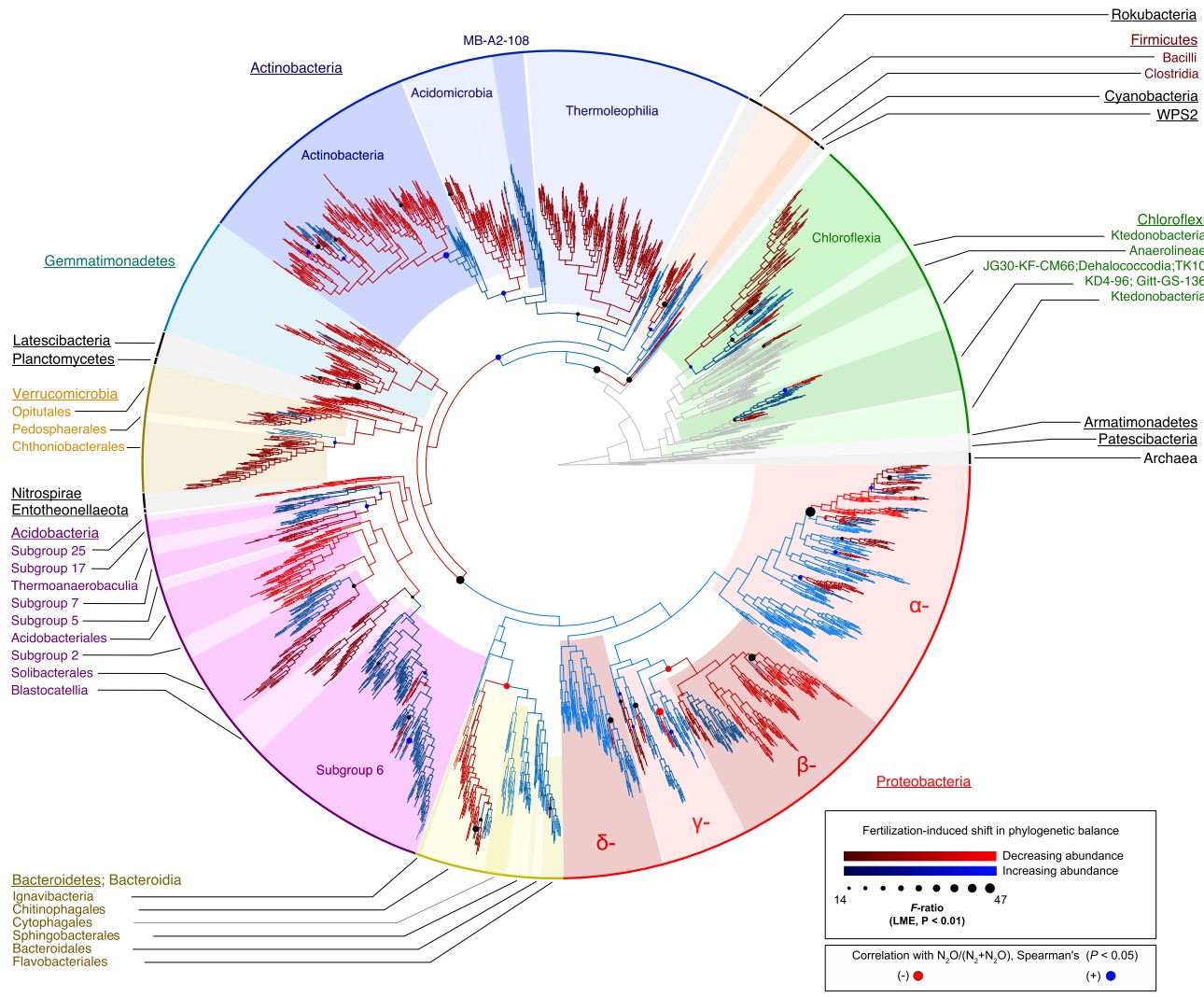

**Fig. 2 Shifts in the phylogenetic composition of communities in response to long-term N fertilization across field experiments, and their relationship with denitrification end-product ratios ($N_2O/N_2O + N_2$).** Significant shifts (false discovery rate corrected $Pr(F) < 0.01$; $N = 105$ independent samples) in the balances of neighbouring clades as a result of fertilization are indicated by circles at respective nodes in the phylogeny. Branch colour denotes clades in each balance that have either increased (blue) or decreased (red) in response to fertilization. Node symbol size and branch colour hue reflect the strength of the fertilization effect, based on $F$-ratios obtained from linear mixed-effects models. Node symbol colour indicates whether changes in the balances are significantly associated (Spearman's $\rho$, $P < 0.05$; $N = 105$ independent samples) with increased $N_2O/N_2O + N_2$ ratios (blue) or decreased ratios (red), or were not significantly associated (black).

less important roles in the evolution of different steps in the pathway[38].

Comparison of denitrification end-product ratios with fertilization-induced shifts in phylogenetic community composition revealed links between increased $N_2O$ emission potential and altered microbial community structure (Fig. 2). The increase in Actinobacteria relative to other bacterial lineages was significantly associated with increased $N_2O/(N_2 + N_2O)$, which is notable as genome-sequenced Actinobacteria with dentrification genes have been shown to have a truncated pathway in which the *nosZ* gene is lacking[12]. However, these associations with increased end-product ratios were specific to relative proportions of certain Actinobacterial lineages, particularly the increased abundance of lineages within the class Actinobacteria relative to the Acidomicrobia in fertilized soils. In contrast, fertilization increased the orders Bacteroidetes, Sphingobacterales, Cytophagales and Flavobaceriales within the Bacteroidetes, and this shift was significantly associated with decreasing end-product ratio. This relationship may be explained by the fact that organisms within

each group, except Cytophagales, are more likely to be non-denitrifying $N_2O$ reducers that possess a clade II *nosZ*, which have been shown to be capable of $N_2O$ consumption in pure culture studies as well as in soil microcosms[12,13,17]. Our results show that fertilization effects are complex and N addition can impact both putative $N_2O$ producers and consumers positively, but that the net effect of the altered phylogenetic community shifts is increased $N_2O$ emission potential.

**Addition of $N_r$ increases resource availability and complexity of soil microbial networks.** Like other N-transformation processes in soils, organisms that perform denitrification often do not have the full repertoire of genes required for the entire pathway, i.e reduction of soluble $NO_3^-$ completely to $N_2$ in denitrification, and thus the different steps of the denitrification pathway can be performed by a complex network of microbial species[39]. Thus, the increase in both C and N related resources caused by long-term addition of $N_r$ may indirectly affect the microbial controls of $N_2O$ emissions by altering microbial

co-associations. This is particularly relevant given that many denitrifiers produce $N_2O$ as a terminal product, whereas others perform only $N_2O$ reduction. However, the directional effect of increased resource availability on the complexity of microbial co-associations in soil habitats is unclear, as studies have shown both increased[20,40] and decreased[41] complexity in response to elevated availabilities of C and N. We therefore compared separate networks of frequently occurring OTUs in unfertilized and fertilized soils with edaphic factors as well as biotic controls of $N_2O$.

Both networks consisted of similar numbers of OTUs and were dominated by positive associations (Fig. 3a). However, the fertilized network was more complex, with three and four times the number of unique positive and negative edges (respectively) compared to the unfertilized network. The majority of edges unique to the fertilized network linked Actinobacterial and Proteobacterial OTUs with those of Proteobacteria, Acidobacteria, and several other phyla, corresponding to reports showing increased co-associations of these groups in soils with elevated C or N availability[20,40]. A large proportion of co-associations of Acidobacteria in the fertilized network were, however, constrained amongst OTUs within this phylum, which likely reflects shared niche preference amongst lineages of this group in the fertilized soils. By contrast, the unfertilized soils showed few dominant edges, i.e. a more even distribution of edges across the taxonomic groups. Comparison of networks of positive associations further showed that long-term addition of $N_r$ resulted in closer connections between microorganisms (Table 1). Clustering coefficient and average connectivity of the fertilized network were higher than that of the unfertilized network, whereas network diameter and average path length were higher in the unfertilized network (Table 1), with all values significantly greater than those generated from random networks ($P < 0.001$). These results suggest that the addition of $N_r$ results in more complex networks in which organisms are more connected, forming fewer disparate communities as indicated by the lower modularity in the fertilized network. Furthermore, increases in both node connectance and Jaccard similarity in the fertilized network indicate a higher degree of ecological overlap amongst OTUs in fertilized soils. Increased complexity and ecological overlap can arise from multiple intersecting mechanisms, including increased cross-feeding or other facilitating interactions, as well as increases in spatial or niche overlap accommodating a wider range of organisms. Theory predicts that higher levels of niche overlap amongst species results in more aggregated patterns of species co-associations[42], suggesting that $N_r$-induced increases in network complexity are largely due to increased realised niches of organisms when resource availability is higher. At the same time, the decreased path length also supports a potential increase in mutualistic interactions, such as cross-feeding of C or N substrates, as the inverse of this metric indicates a higher overall efficiency in the system[43]. Furthermore, higher resource levels in the fertilized soils should also activate fast-growing microorganisms favoured by easily available resources and thereby increase competition, which is supported by the higher number of negative associations in the fertilized than the unfertilized soil (Fig. 3a). Although network analyses cannot define the mechanisms, our results suggest that increased resources increases the degree of complexity in microbial networks, similar to what was shown in soil subjected to experimental warming[44].

We compared the topologies of the two networks using the DyNet network tool, which identifies the nodes and linkages that are shared between two or more networks, as well as those that are unique to each network. This analysis showed substantial restructuring of communities due to fertilization, and distinct modules (minimum five OTUs) were identified in each network (Fig. 3b). These modules can be regarded as sub-communities that are associated by shared niche space and putative biotic interactions[45] and will hereafter be referred to as communities. The four largest communities in both unfertilized and fertilized networks shared similar subsets of OTUs, but either gained or lost OTUs that were not members of other detected communities depending on fertilization (Fig. 3b). Communities A and B consisted of a diverse range of bacterial phyla, although there was no pattern amongst taxa that were common or unique between unfertilized and fertilized networks (Supplementary Fig. 2). Both communities increased in complexity in the fertilized network, with the addition of new OTUs as well as rewiring of nodes common to both networks. By contrast, communities C and D were more complex in the unfertilized soils, and had reduced numbers of OTUs and co-associations in fertilized soils. Communities that were unique to each network also included OTUs that varied in community membership between unfertilized and fertilized soils. For example, uA and uC were unique to the unfertilized network and included OTUs that were present in both networks, but restructured such that they did not form identifiable communities in fertilized soils.

**Restructured networks differed in niche space and affected denitrification functionality**. Comparison of the abundances of communities identified in the networks to edaphic factors was performed by calculating community module eigenvectors, which collectively represents overall abundances of OTUs within each community[40]. We found that changes in niche space were affected by increased $N_r$ across communities common to each network (Fig. 3c). For example, the abundance of community A was unaffected by soil pH in unfertilized soils, yet increased in abundance with decreasing soil pH and increasing C/N ratio in the fertilized soils. By contrast, abundances of C and D were correlated with similar edaphic factors, although differences in variables affected by N fertilization, such as $NO_3^-$, P, and C/N, were observed (Fig. 3c; Supplementary Table 3). The communities unique to fertilized soils had more significant and stronger correlations with edaphic factors than those unique to the unfertilized soils, indicating that the addition of $N_r$ increased the importance of environmental filtering of associations within communities.

We then assessed the implications of $N_r$ induced complexity and rewiring of microbial networks on microbial controls of $N_2O$ emissions. Comparison of gene abundances and activities with community abundances support that rewiring by long-term $N_r$ addition reflects changes in denitrification functionality (Fig. 3c). Abundances of four communities that were unique to the unfertilized network were positively correlated with *nosZ/nir* gene ratios, two of which were also negatively correlated to the denitrification end-product ratio. Also, fertilization resulted in community A being positively associated with the end-product ratio, with corresponding decreases in *nirS* and *nosZ* clade I gene abundances. Similarly, the abundance of community B was associated with increased $N_2O$ production rate and end-product ratio in both soils, however this relationship become stronger in the fertilized soils and its abundance was negatively correlated with *nosZ* clade I and *nirK* abundances. Abundances of both A and B increased with pH regardless of fertilization, and corresponded to the negative correlations with end-product ratio Overall, this highlights that altered denitrification and $N_2O$-reducing functionality can be linked to $N_r$-induced reorganization of distinct microbial communities that occupy different niches in soil habitats. Future research on how resource availability shapes the interactions between well-defined functional groups may provide additional insight into the mechanisms that link community composition and $N_2O$ functioning.

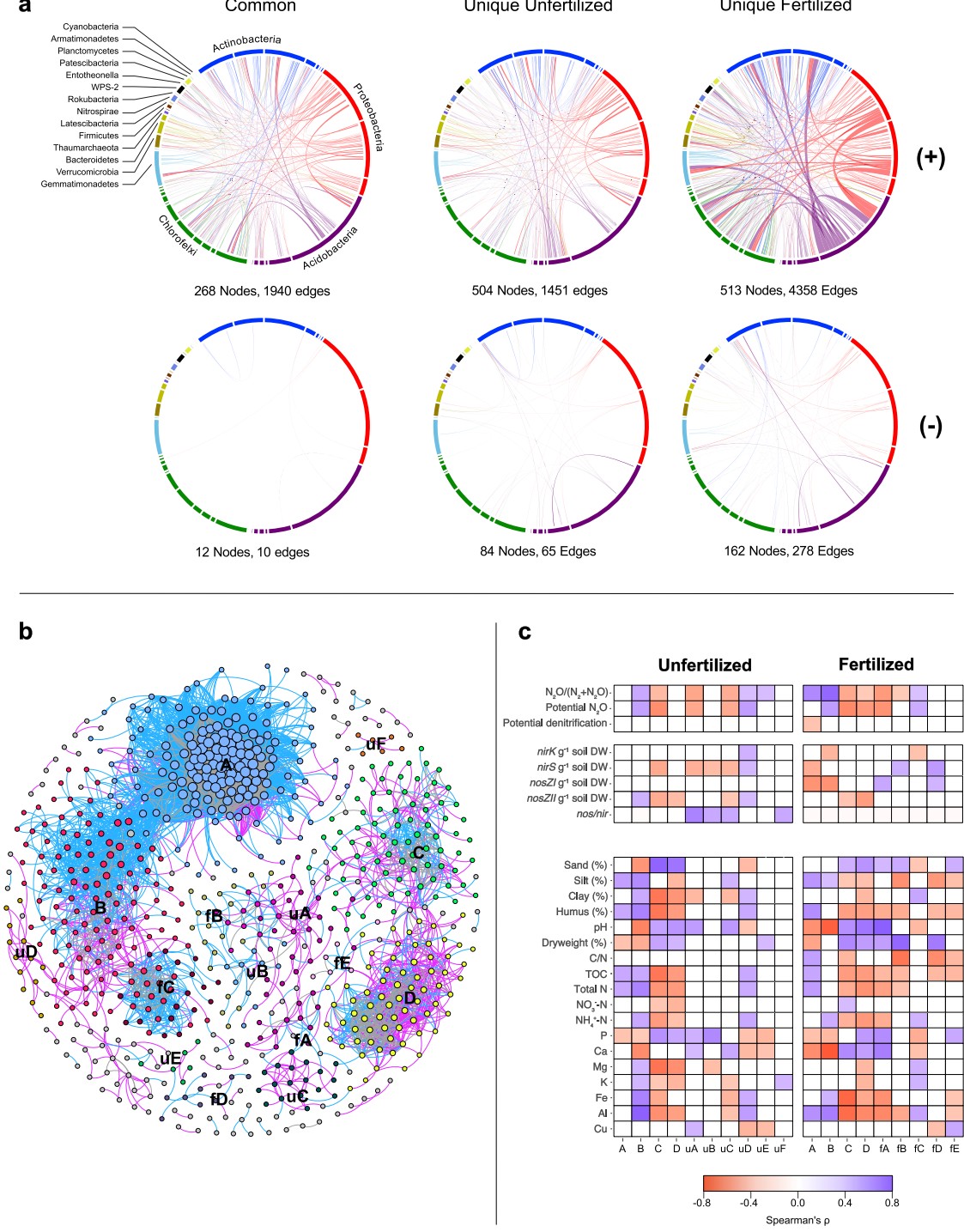

**Fig. 3 Co-association networks of OTUs in fertilized and unfertilized soils. a** Taxonomic groups at the class level are represented by coloured segments in the outer ring, and ribbons represent significant co-association (+) or exclusionary (−) relationships between the different taxonomic groups. The width of the ribbons is proportional to the number of links between the OTUs within each segment, while colour indicates which segment of the two has a higher number of total links. Note that the number of nodes in common and unique subgraphs may overlap; only the number of edges are non-overlapping. **b** Comparison of fertilized and unfertilized co-occurrence network topologies by network alignment. Nodes are grouped into modules detected in both fertilized and unfertilized networks, where node colour indicates module membership and edge colour corresponds to co-associations that are unique to unfertilized (pink) or fertilized networks (blue), or common to both networks (grey). **c** Heatmaps show the correlation of module eigenvalues in each network with potential activities and abundances of denitrifying and N$_2$O reducing communities, as well as soil edaphic factors. Tile colour reflects the strength and direction (blue = positive, red = negative) of correlations, and non-significant correlations (Spearman's ρ, P > 0.05; N = 51 and N = 54 independent samples for unfertilized and fertilized soils, respectively) are left blank.

**Table 1 Topological properties of microbial co-association networks in unfertilized and fertilized soils.**

| Network | $r_T$[1] | Network Size | | Network Diameter | Ave. path length | Average connectivity | Clustering coefficient | Modularity | Connectance | Average Jaccard similarity |
|---|---|---|---|---|---|---|---|---|---|---|
| | | Nodes | Edges | | | | | | | |
| Unfertilized | 0.85 | 546 | 3391 | 19 | 5.977 | 12.75 | 0.563 | 0.56 | 0.024 | 0.021 |
| Fertilized | 0.88 | 532 | 6298 | 11 | 3.603 | 23.07 | 0.672 | 0.36 | 0.042 | 0.035 |

[1]Threshold of Pearson's correlation coefficient determined for each network using random matrix theory (RMT).

**Relative importance of biotic and abiotic controls of $N_2O$ emissions**. To determine the importance of controlling variables that determine $N_2O$ emissions from soils, we generated separate machine learning-based models for unfertilized and fertilized soils and examined the relative importance of biotic and abiotic factors in predicting denitrification end-product ratios. Changes in community structure were included as abundances of community modules identified in each network. The most important predictor variables (median relative influence > 5%) in unfertilized soils were all biotic variables, whereas long-term addition of $N_r$ increased the relative importance of abiotic variables (i.e. pH and Ca content, Fig. 4). Similar to Samad et al. (2016)[46], increased community diversity based on Shannon's $H'$ corresponded with a decreased end-product ratio in the unfertilized soils, yet was not an important predictor variable in fertilized soils. Overall abundances of the communities A, uA and uC, as well as the abundance and diversity of total microbial communities (abundance of the 16S rRNA gene, Supplementary Table 4) were the most important predictors of end-product ratios in unfertilized soils, and increases in each of the important biotic variables, except community A, corresponded to decreased end-product ratio (Fig. 4a).

In fertilized soils, soil pH was the most important predictor of end-product ratio and the second most important abiotic predictor was Ca content. While fertilized and unfertilized soils did not differ regarding pH, soil Ca concentration was ~14% higher in the fertilized soil (Fig. 4b, Supplementary Table 1). Soil acidity is a strong driver of $N_2O$ emissions[24], and global soil $N_2O$ emissions have been shown to be more sensitive to changes in pH in fertilized soils although the underlying mechanism is unclear[47]. Moreover, microbial community composition and diversity can modulate the effect of soil pH on $N_2O$ emission potential[48,49]. Among the biotic variables, communities B and C, common in both unfertilized and fertilized soils, were important predictors of end-product ratio in fertilized soils, with communities B and C being associated with increasing and decreasing $N_2O$ production, respectively. The model further indicates that increased abundance of *nosZ* clade II is associated with decreased end-product ratio in fertilized soils (Fig. 4b), although the abundance of *nosZ* clade II only increased by 13% in the fertilized soils (P = 0.06; Supplementary Table 4). This is similar to a recent report showing that the abundance of *nosZ* clade II $N_2O$ reducers increased after fertilization[50] and agrees with previous work on the importance of *nosZ* clade II for greater $N_2O$ sink capacity in agricultural soils[51]. Among the other important functional predictors of the end-product ratio, the ratio of *nosZ* to *nir* gene abundance showed decreasing ratios with increasing *nosZ* to *nir* gene-abundance ratios (ANCOVA $F_{1,103} = 22.33$, P < 0.001; standardized regression coefficient = −0.31). While this relationship was not affected by fertilization ($F_{1,103} = 2.44$, P = 0.12) and the ratio of total *nosZ* to *nir* gene abundance did not differ between unfertilized and fertilized soils (Supplementary Table 4), the accumulated local effects curve, which shows the relationship between *nosZ/nir* and end-product ratios in isolation from other predictor variables in the model, in the fertilized soils indicates a

threshold in this relationship. This shows that increasing *nosZ* to *nir* abundance ratios higher than the threshold level has no effect on the end-product ratio. This threshold effect as well as the overall differences in variable importance between fertilized and unfertilized soils suggest that $N_2O$ production is less tightly regulated by microbial communities in soils with elevated resources and thereby abiotic controls, in particular the effect of soil pH, become more important. Previous work has shown that denitrification activity in soil is dependent on bacterial community composition, whereas broadly defined microbial functions, such as respiration, are driven primarily by resource availability or other edaphic factors[52,53]. However, similar to Philippot et al.[52], our results suggest that the relationship between denitrification functionality and microbial community structure is modified by the addition of resources. In this case, long-term addition of $N_r$ resulted in significantly restructured communities, yet weakened microbial controls on potential $N_2O$ emissions. This is further supported by the stronger environmental filtering of communities identified in the network in fertilized soils, yet greater importance of soil pH and Ca in predicting the ratio of denitrification end products.

In conclusion, by leveraging multiple field experiments from sites with varying soil types and microbiomes, we show that long-term addition of $N_r$ has a generalized effect on the phylogenetic structure of microbial communities that are linked to increased $N_2O$ emission potential. Although the long-term field sites surveyed in this study were all located within Sweden, they were dispersed largely north to south over ~138,000 $km^2$ that included three defined climate zones also found in continental Europe, eastern and midwestern North America and smaller regions in eastern Asia and South America[54]. Furthermore, the greatest increases in N fertilizer usage over the past 60 years has occurred in regions within similar climate zones[55]. Nevertheless, it may be difficult to extrapolate these findings to warmer tropical and arid climates, or arctic and alpine regions. In addition to also altering the direct genetic controls of net $N_2O$ production, $N_r$ has a homogenizing effect on microbial communities such that organisms are more closely linked through a combination of increased potential interactions and a higher degree of shared niche space. Biological controls of $N_2O$ emissions were more important in unfertilized soils, and we show that N addition increases the relative importance of abiotic soil factors in determining potential $N_2O$ emissions from arable soils. We propose that this shift towards greater importance of abiotic controls reflects an overall weakening of direct microbial regulation of $N_2O$ emissions due to increasing resource levels in soil (Fig. 5). This is a potential mechanism underlying increased $N_2O$ emissions with increasing $N_r$ levels, where N addition causes positive feedback that creates a negative spiral with increasing $N_2O$ emissions. Our findings have ramifications for predicting the consequences of long-term addition of $N_r$ on future $N_2O$ emission rates from agricultural soils, as both the geochemical legacy of long-term $N_r$ addition[56], as well as an inherent 'microbial legacy' that determines the response of $N_2O$ emissions to differences in edaphic factors need to be considered.

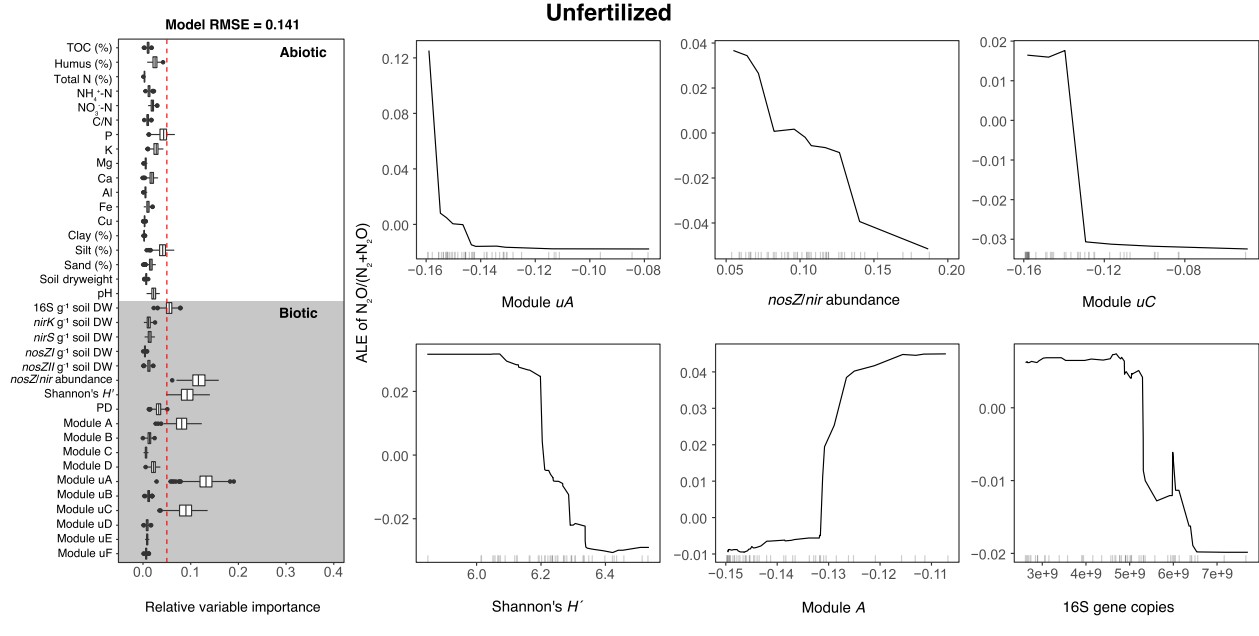

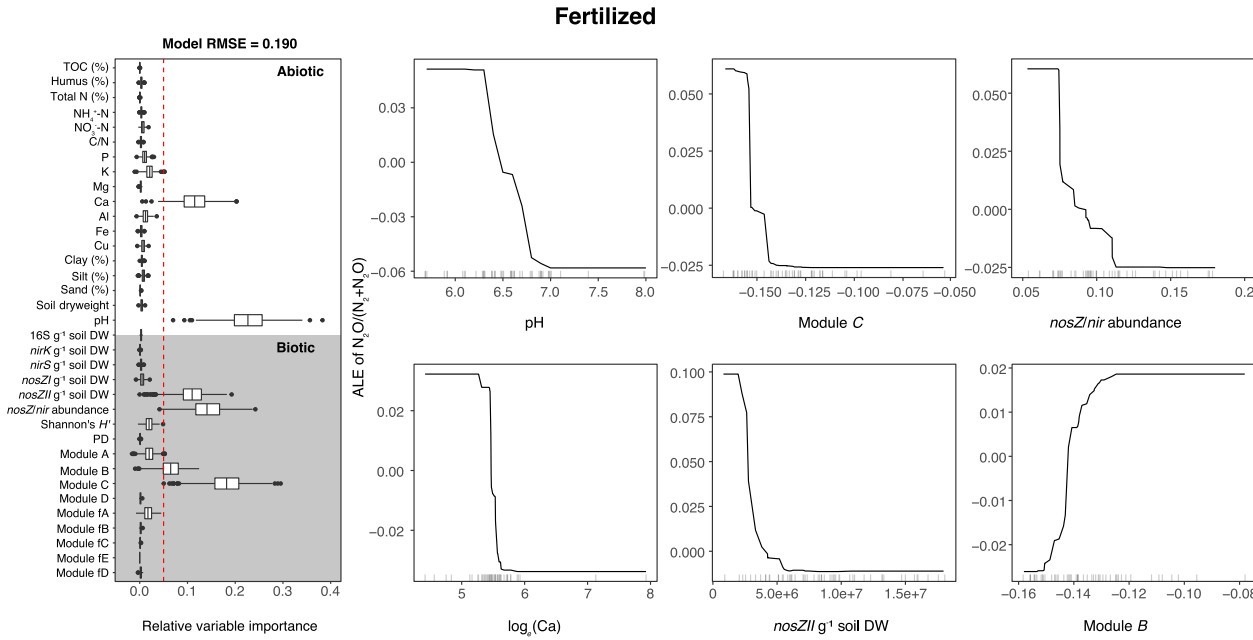

**Fig. 4 Relative importance of abiotic and biotic factors in predicting denitrification end-product ratios (N₂O/[N₂ + N₂O]) in fertilized and unfertilized soils based on generalized boosted regression modelling.** Model fit is indicated by the residual mean squared error (RMSE) in the figure. Predictor variables with median relative permutation importance ($n = 500$ permutations) >5% were used to generate accumulated local effects (ALE) plots, which show the relationship between the predictor variables (x-axis) and end-product ratios in the model (y-axis) while accounting for potential correlations amongst predictor values. Note that the scale of the end-product response in the y-axis is normalized in ALE plots based on the conditional response within a range of the predictor value. For boxplots, box limits represent the inter-quartile range (IQR) with median values represented by the centreline. Whiskers represent values ≤1.5 times the upper and lower quartiles, while points indicate values outside this range.

## Methods

**Soil sampling and analyses of soil properties**. Soil samples were taken in October and November 2013 from 14 long-term field trials located in different regions across Sweden (Supplementary Table 1). Each field trial included unfertilized and mineral fertilization treatments ranging from 80–150 kg N ha⁻¹ year⁻¹ that have been managed for a minimum of 15 up to 57 years under annual cereal crop rotations. All trials consisted of 2–6 field replicates per treatment, and soil was collected from each replicate plot for a total of 108 samples. For each sample, five cores from the topsoil (0–20 cm) were taken from each field plot, then

homogenised by sieving through a 4 mm mesh. A subsample was used for a physico–chemical analysis (Agrilab AB, Uppsala, Sweden; Supplementary Table 2), and the rest stored at −20 °C for later processing.

**Potential denitrification and N₂O production**. For each sample, two portions of 10 g fresh weight soil were each weighed into 125 mL Duran bottles and made into slurries by adding 20 mL distilled water. The bottles were capped and the headspace exchanged by flushing with N₂. For each sample, potential denitrification was measured in one bottle by injecting acetylene to reach a partial pressure of 0.1 atm,

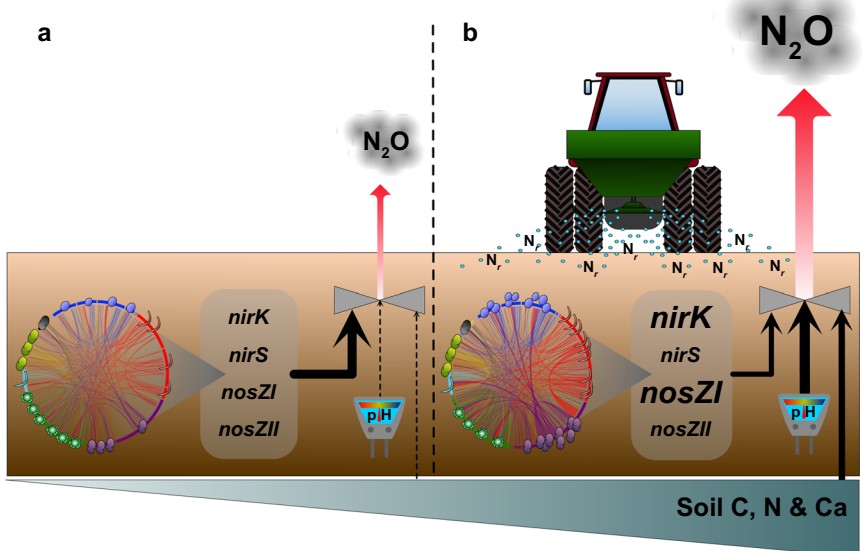

**Fig. 5 Conceptual model of the effect of long-term addition of reactive N on microbial controls of N₂O emissions. a** In unfertilized soils, changes in biotic factors associated with microbial communities, such as their diversity, patterns of co-association and abundances of functional genes, exert stronger control over potential N₂O production compared to abiotic factors including soil pH and resource levels. **b** Long-term addition of reactive N through fertilization restructures microbial communities over time, resulting in increased complexity of microbial co-association networks as well as altering denitrification functionality through increased abundances of *nirK* and *nosZ* clade I genes associated with denitrification and N₂O reduction. However, abiotic soil factors, especially soil pH, become more important in determining potential N₂O emission than changes in microbial communities.

whereas potential N₂O production was measured in the other bottle by not adding acetylene. After 0.5 h of pre-incubation at 25 °C with agitation (175 rpm), 1 ml of substrate was injected into each bottle to reach a final concentration of 3 mM KNO₃, 1.5 mM succinate, 1 mM glucose and 3 mM acetate. Gas samples were taken every 0.5 h for a total duration of 2.5 h and N₂O concentration was determined using a gas chromatograph (Clarus 500, Elite-Q PLOT phase capillary column; Perkin Elmer, Hägersten, Sweden). The rate of N₂O accumulation in each bottle was determined by non-linear regression, and the denitrification end-product ratio of each soil was calculated as the ratio of potential N₂O production rate (without acetylene) to the potential denitrification rate (with acetylene).

**Extraction of DNA and quantification of 16S rRNA and N-cycling functional marker genes**. DNA was extracted from 300 mg of each soil using the FastDNA kit (MP biomedicals, Santa Ana, CA USA) following manufacturer's instruction, then quantified using a Qubit fluorimeter (Invitrogen, Sweden). Real-time quantitative PCR of the 16S rRNA gene and denitrification genes was then performed to estimate quantities of the microbial community as well as targeted functional groups. Prior to quantification, inhibition tests were performed for all samples by adding a known amount of pGEM-T plasmid to 10 ng of extracted soil DNA or water, followed by real-time quantitative PCR using plasmid-specific primers T7 and SP6. No inhibition of the PCR reactions was detected based comparison of cycle threshold ($C_t$) values between DNA extracts and water-only controls. Primer combinations and thermal cycling conditions used to quantify 16S rRNA and functional genes are described in Supplementary Table 6, and all reactions contained iQ™ SYBR Green Supermix (Bio-Rad, Hercules CA, USA), 0.1% bovine serum albumin (BSA; New England Biolabs, Ipswich MA, USA) and between 5–10 ng DNA. Final primer concentrations varied between 0.5 µM for 16S rRNA and *nosZ*II, to 1 µM for *nosZ*I, *nirK* and *nirS*. Each gene was quantified in duplicate 15 µl reactions and the qPCR efficiencies ranged from 73 to 98%.

**Sequencing and analyses of total bacterial and archaeal communities**.
Amplicons of the V3-V4 region of the 16S rRNA gene were prepared following a two-step procedure. The first step PCR reactions consisted of 1× Phusion PCR Mastermix (Thermo-Fisher scientific, Stockholm, Sweden), 1 mg/ml BSA and 0.25 µM of primers Pro341 and Pro805r[57]. Duplicate 15 µl reactions were performed for each sample using the following thermal cycling conditions: an initial denaturing step of 3 min at 98 °C, followed by 25 cycles of 98 °C for 30 s, 55 °C for 30 s and 72 °C for 30 s, then a final extension step of 10 min at 72 °C. Resulting PCR products were then pooled and purified using HighPrep PCR Clean-up beads (MagBio Genomics, Gaithersburg, MD, USA) following the manufacturer's protocol. Barcodes were then added in the second PCR step using 0.2 µM Nextera barcoded primers (Illumina, San Diego CA, USA) and 15% of the purified PCR product from step 1 PCR. Reactions were performed in duplicates of 30 µl and thermal cycling conditions remained the same as the first step except 8 cycles were performed and the extension step at 72 °C was prolonged to 45 s. Sequencing was

performed by Microsynth (Balgach, Switzerland) on a MiSeq Illumina sequencer using V2 2 × 250 paired-end chemistry.

Obtained paired-end reads were trimmed using the FASTX-toolkit (http://hann onlab.cshl.edu/fastx_toolkit), merged using PEAR[58] (minimum overlap = 20 bp, minimum quality score = 30, minimum and maximum merged read-lengths of 300 and 505, respectively) and quality filtered using VSEARCH[59] such that merged reads with the maximum expected error above one were discarded. Following paired-read merging and quality filtering, 4,261,303 reads were retained for further processing. Reads were then dereplicated and clustered into OTUs using VSEARCH with a minimum sequence similarity of 0.98. Chimaeras were removed using de-novo chimaera detection in combination with reference-based chimaera checking using 16S rRNA sequences using the SILVA database (release 132) as the reference database. Representative OTU sequences of the resulting 3643 clusters were aligned and classified using the SINA algorithm[60] with the SILVA database as a reference. The alignment was manually inspected with the ARB software[61] and OTUs identified as chloroplasts or mitochondria were removed. All sequence data is available at the NCBI Short Read Archive under BioProject accession PRJNA722868.

**Analysis of community diversity and structure**. Communities were partitioned into 'frequent' and 'rare' OTUs by examining species abundance distributions for each dataset (Supplementary Fig. 3). For all OTUs, the index of dispersion ($I$) was calculated as the ratio of the variance in abundance across all samples to the mean abundance, multiplied by site occupancy[33]. Frequent community OTUs were identified as those for which $I$ deviated significantly from a $\chi^2$ distribution ($\Pr(I) < 0.05$), resulting in species abundance distributions that follow a log-normal distribution. All calculations were based on the mean abundances of OTUs obtained from 100 instances of rarefied OTU tables.

The diversities of frequent total prokaryotic communities were then calculated as Shannon's index, species richness and Phylogenetic Diversity (PD) using the 'vegan', 'phyloseq' and 'picante' packages[62–64]. To assess the effect of long-term fertilization on the structure of total prokaryotic communities, non-rarefied tables of frequent OTUs were initially transformed using the phylogenetic isometric log-ratio transformation (PhILR[32]). This method accounts for the compositional nature of microbial community data and results in a matrix of samples and 'balances', where each balance is associated with a node in the OTU phylogeny. Values for each sample are calculated as the log-ratio of the abundances of taxa descending from either side of the node, where positive values indicate higher abundances of taxa in the numerator relative to the denominator, while negative values indicate the reverse. A pseudocount was added to all zero values using the Bayesian zero-imputation method implemented in the 'zCompositions' package in R[65]. Following PhILR transformation, community compositions were examined using Euclidian distances followed by non-metric multidimensional scaling. Significant shifts in community composition in response to fertilization were tested for using permutational ANOVA (PermANOVA) implemented in the 'adonis2' function of the 'vegan' package in R, with permutations ($n = 1000$) restricted to

within sampling sites using the "strata" function. To examine the effect of fertilization on phylogenetic balances, linear mixed-effects modelling was performed for each balance such that sample location and fertilization were treated as random and fixed factors, respectively. The significance of the fertilization term was assessed by model comparison in which the fertilization term was excluded, and tests of $F$-ratios were performed using the Kenward–Roger approximation of degrees of freedom. Balances exhibiting a significant response to fertilization ($Pr(F) < 0.01$ after correction for false discovery rate) were retained, and mean values of the balances in fertilized and unfertilized plots were estimated based on model results using the 'lmerTest' package in R[66].

**Co-association network analyses and module detection in unfertilized and fertilized soils**. Networks of frequently occurring OTUs were inferred separately for fertilized and unfertilized plots using the 'igraph' package in R[67]. Non-rarefied matrices of frequently occurring OTUs were initially transformed using the centred log-ratio transformation to account for compositionality in the datasets, followed by calculation of Pearson correlations between each pair of transformed OTU abundances within unfertilized or fertilized plots across locations. Final networks were then inferred by random matrix theory using the 'RMThreshold' package in R[68]. This method identifies thresholds of correlation coefficients based on the transition of the empirical nearest-neighbour eigenvalue spacing distribution (NNSD) from Gaussian orthogonal ensemble to a Poisson (or exponential) distribution, indicating the point at which the inherent structure of the network is separated from noise. Since each dataset may contain different levels of noise, the selection of thresholds was based on Kolmogorov–Smirnov tests of the empirical NNSD distributions to the theoretical exponential distribution (Supplementary Fig. 4). We selected the first threshold value that was non-significant ($P > 0.05$) for each network, indicating the NNSD had transitioned to an exponential distribution and thus 'noise' linkages specific to each network were removed. Plots of the sum of squared errors between the empirical NNSD and the exponential distribution over the range of tested thresholds were also examined to confirm the goodness of fit, and are shown in Supplementary Fig. 4. This resulted in threshold values of Pearson's $r = 0.85$ and $r = 0.88$ for unfertilized and fertilized networks, respectively. All remaining correlations were highly significant (false discovery rate corrected $p < 0.001$), and isolated nodes with degree = 0 were removed.

The structures of unfertilized and fertilized networks were then compared by identifying edges between OTUs that were either common or unique to each fertilization level, and subgraphs of each edge set were extracted and visualized in CIRCOS plots using the 'circlize' package in R[69]. Differences in the topologies of co-association (i.e positive edges only) networks were then detected using DyNet[70], which identifies changes in linkages amongst nodes between two or more networks. We then used 'igraph' to calculate various metrics to describe network complexity, such as average network connectedness (average node degree), network transitivity (clustering coefficient), average path length and modularity. Furthermore, node connectance and average Jaccard similarity per node were calculated to determine the degree of ecological overlap amongst OTUs in each network. The significance of graph metrics was determined by generating 1000 random networks with the same number of nodes and edges as the fertilized and unfertilized networks using the Erdos–Renyi model, and probability values were determined using two-tailed tests of the observed metric values compared to the distribution of random values. Finally, relationships between biotic or abiotic factors and the topologies of each network were performed in the same manner as outlined in Jones and Hallin (2019)[71]. Briefly, the 'edge betweenness' algorithm[72] was used to identify distinct modules of co-occurring core OTUs in either unfertilized or fertilized soils. Modules consisting of more than five nodes were then used in eigengene analysis, in which a single eigenvector reflects the overall change in abundance OTUs in modules across samples[73]. Across all modules, the variance explained by each eigenvector ranged from 52 to 91%. The resulting eigenvalues were then compared to abiotic and biotic factors by Spearman correlations.

**Generalized boosted regression modelling and variable importance**. The relative importance of different biotic and abiotic factors in predicting denitrification end-product ratios in unfertilized and fertilized soils was determined using generalized boosted regression modelling. This method allows for modelling of non-linear relationships between predictor and response variables, while also dealing with issues of non-normality and collinearity amongst predictor variables[74,75]. To avoid overfitting the model, algorithm tuning was performed via a grid search approach using the 'caret' package[76] to obtain the optimal number of trees, shrinkage parameter, interaction depth and a minimum number of observations in tree nodes for each dataset. Model validation was performed by ten-fold cross-validation, and relative variable importance was then determined by permutation variable importance with 500 permutations using the 'vip' package[77]. Accumulated local effects plots implemented in the 'iml' package[78] were used with a grid size of 10 to assess how the top predictor variables (relative importance > 5%) are related to denitrification end-product ratios in each dataset.

**Statistics and reproducibility**. All statistical analyses were performed using R as stated in the descriptions of each individual analysis. Replicates within each combination of the field site and fertilization treatment are defined as individual

field plots, and the number of plots per treatment per site are given in Supplemental Table S1.

**Reporting summary**. Further information on research design is available in the Nature Research Reporting Summary linked to this article.

## Data availability
Sequence data is available from the Short Read Archive at the National Center for Biotechnology Information (NCBI) under BioProject accession PRJNA722868, and all relevant soil, gene abundance and filtered OTU data are provided in a separate excel file as Supplemental Data.

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

## Acknowledgements
This work was supported by the European Union (Marie Curie ITN NORA, FP7-316472) and the Swedish Research Council (grant 2016-03551).

## Author contributions
S.H. and C.M.J. designed the study, M.P. and M.T. performed sampling and laboratory analyses, C.M.J. and M.P. performed the statistical analyses and C.M.J., M.P. and S.H. contributed to the writing of the manuscript.

## Funding

## Competing interests
The authors declare no competing interests.
