## [Peer Review File · Communications Biology]

reviewers' comments:

Reviewer #1 (Remarks to the Author):

Review

Here the authors present a thorough study of Nr effects on microbial communities and relate changes to denitrification products. Overall, the study is sound and adds an important component to our understanding of increasing N additions to soils.

My overall concern is that this study is only on soils in Sweden, so while there is quite a bit of variation in soil types, there is not variation in climatic factors that could have an additional effect on results – this needs to be addressed and specifically discuss how these results can be extrapolated out to other regions (or if not possible, then this should be made clear).

I also think a conceptual model would really help bring this manuscript to the next level: it would show how N addition affects denitrification production (+/-), microbial functional composition, and abundance of genes involved in denitrification and compare fertilized vs unfertilized soils.

Finally more clarity is need around the balance between abiotic and biotic controls (rewires and relaxes are not very clear terms to use throughout).

Abstract:

Novelty statement is needed, make clear what is special about this research and how it is different from other studies of N addition. What is this adding to the literature? The beginning starts strong but a sentence or two at the end is needed.

Introduction

L41-43: clarify this important sentence: in other words, is the increase a result of abiotic or biotic

L67-68: Elaborate on this, it is critical to the hypothesis of the study but needs more explanation. Especially make clear why network is the analysis to use to determine this.

L74-75: add in key info to the main text: average duration of N addition, location (temporal sites?), N type added, etc.

93: instead of 'relaxes' can you say something like - decreases the relative ratio between biotic to abiotic controls

Results and Discussion

L135: small r

L152: or they are outcompeted, but this would not necessarily mean they are related ecologically

L166: but horizontal gene transfer was not tested here so don't over state.

Methods

Likely some text can go in supplement, as is, is very long.

Figure: Would it be possible to add * next to clades that have known potential to have N related genes (from literature)

Figure 4: ALE plots could be moved to supplement

Reviewer #2 (Remarks to the Author):

This paper used an impressive dataset to demonstrate legacy effects of Nr on N₂O emission via microbial structure. The authors collected 108 soil samples from 14 long-term fertilization field experiments and used qPCR & sequencing to investigate the genes relating to N₂O emission, and the microbial clustering driving N₂O emission via network analysis. They showed that Nr legacy altered the relative importance of biotic and abiotic predictors on potential N₂O emissions, and biotic factors are more important in unfertilized soils. The bioinformatic and statistical analysis were well performed. I've a few comments below.

(1) N₂O from agricultural soils can also, as a byproduct, come from the ammonia oxidation process. In this paper, only the denitrification genes were investigated, while the amoA was not referred to. It might not be necessary to do extra work in the lab; but given the authors had sequencing data of 16S rRNA gene, is it possible to show a little bit relating result via the ammonia oxidizers in Beta-proteobacteria group/Thaumarchaeota?

(2) The network analysis part is fantastic but also a bit confusing to me on two points. (a) Please unify the network name as co-occurrence network or co-association network. Given OTUs abundance were used, I think co-association is a more accurate term. (b) Fig3a: The colored segment, i.e., the taxonomic groups are not presented in every group, e.g., in common (-) network, only seven lines present while most of the taxonomic groups do not present here. If possible, it might be better to remove those non-related color segments from each network. Just FYI.

Fig1: please describe all significant levels in the legend (the one of NirS.)

Reviewer #3 (Remarks to the Author):

The manuscript "Reactive nitrogen rewires and relaxes microbial controls of soil N₂O emissions" reported effects of Nr addition on soil N₂O production by analyzing soil denitrification potential, microbial community composition, and denitrification related functional gene abundance in 14 field trials with long term fertilization. The authors reported increased complexity of data association-based microbial networks under fertilization. They found that edaphic factors pH and Ca became important predictors of denitrification end product ratio in addition to biotic factors in fertilized soil. The central question discussed in this manuscript is interesting and important to the prediction of agricultural impact on climate change. The experiments and analysis performed are comprehensive in addressing the scientific question discussed. However, I found the wording at many places unclear and affects reading and understanding (see below detailed comments). In addition, since association network has an intrinsic problem in telling mechanisms underlying the links, results regarding the increased network complexity under fertilization, and biological meaning of separating communities within networks are descriptive. I hereby suggest revising the manuscript for clarity and accuracy.

Detailed comments:

Title. I had a hard time interpreting "rewires microbial controls" before I read about the networks. I think the authors meant that the "links were rewired among nodes". "Rewire" is not informative here in the title. From the results, I see abiotic factors pH and Ca became important predictors of denitrification end product ratio. However, there were still a few key microbial-related variables. So I think "relax" lacks accuracy.

L62-64. Explain "higher taxonomic groups". I am confused about the reasoning behind this sentence. Are there reports of changes in abundances of organisms under fertilization that do not harbor N-related traits?

L117. abundance of the gene ...

L190. not clear what "in a modular fashion" means

L192. How microbial association could influence N₂O emission? Please elaborate.

L215. "More homogenized" does not mean "more connected". Homogeneous networks refer to networks where all nodes have the same function(i.e., same connectivity). The authors did not report parameters related to network homogeneity.

L222. "increased interactions" refers to what?

L224-225. Generally, more available resource is expected to alleviate competition. Are some of the negative links connecting fast growers in the network for fertilized soils?

L228. Again, "Homogenizing effect" is confusing here.

L229. How does increased complexity point to a more dynamic community?

L232. Please explain "network alignment".

L236-247. This paragraph talked about modules in the networks, but what are the implications? From Figure S4, the common modules A, B, C and D contained many unique nodes, and the unique modules also contained many common nodes. What is the criterion used to define if a module is shared by two networks or not? This poses a problem in the interpretation of results in Figure 3c and Figure 4, especially for the common modules in the two networks since these modules are very different in the two networks.

L267-268. But two of them showed no correlation with N₂O/[N₂+N₂O] ratio in Figure 3c.

L317. Define ALE.

L793. composition of communities....

Figure 2. Add legends to node symbol size and branch color hue. Also, it is hard to identify changed clades. Circles are at the fork and not clear which clade or both clades after the fork changed.

Figure 3. Add a legend for strengths of correlation

Table S1. Fertilization amounts unclear

Figure S4. I don't see grey edges. What are the percentages of the common/unique nodes and links?

Response to reviewers' comments

Reviewer #1 (Remarks to the Author):

Review

Here the authors present a thorough study of Nr effects on microbial communities and relate changes to denitrification products. Overall, the study is sound and adds an important component to our understanding of increasing N additions to soils.

Comment 1. My overall concern is that this study is only on soils in Sweden, so while there is quite a bit of variation in soil types, there is not variation in climatic factors that could have an additional effect on results – this needs to be addressed and specifically discuss how these results can be extrapolated out to other regions (or if not possible, then this should be made clear).

A: We appreciate the reviewer's comments on the importance of variation in climate conditions, and have addressed this in the concluding paragraph of the manuscript (L363-369).

Comment 2. I also think a conceptual model would really help bring this manuscript to the next level: it would show how N addition affects denitrification production (+/-), microbial functional composition, and abundance of genes involved in denitrification and compare fertilized vs unfertilized soils.

A: As suggested, we have added a conceptual figure (Fig. 5, see below) and an accompanying figure legend.

Comment 3. Finally more clarity is need around the balance between abiotic and biotic controls (rewires and relaxes are not very clear terms to use throughout).

A: We understand that these terms did not come across clearly. We have therefore replaced the term 'rewire' with 'restructure' and 'relaxes' with 'weakens' in the title and throughout. We have also made an effort to clarify the term weaken and better describe how it relates to the relative importance of abiotic and biotic controls of denitrification end-product ratios (L345-356).

Abstract:

Comment 4. Novelty statement is needed, make clear what is special about this research and how it is different from other studies of N addition. What is this adding to the literature? The beginning starts strong but a sentence or two at the end is needed.

A: We have adjusted the closing sentence to better reflect the knowledge gap mentioned in line 21-23.

Introduction

Comment 5. L41-43: clarify this important sentence: in other words, is the increase a result of abiotic or biotic

A: We have revised this sentence to clarify the contrast between abiotic and biotic factors (L39-43). If the increase in N₂O as the end-product of denitrification is the result of mainly abiotic or biotic factors is uncertain and that is what we investigate in this study.

Comment 6. L67-68: Elaborate on this, it is critical to the hypothesis of the study but needs more explanation. Especially make clear why network is the analysis to use to determine this.

A: We have elaborated on this by emphasizing that one of the main strengths of network analysis is its ability to represent 'emergent properties'; in this case, denitrification and N₂O reducing functionality that is driven by various taxa within a community (L68-78).

Comment 7. L74-75: add in key info to the main text: average duration of N addition, location (temporal sites?), N type added, etc.

A: This information has now been provided in the revised text (L83-85) and Table S1 shows exact location for each experiment.

Comment 8. 93: instead of 'relaxes' can you say something like - decreases the relative ratio between biotic to abiotic controls

A: Thank you for the suggestion; we have revised as suggested and here also define how biotic controls are weakened by N addition: "Abiotic predictors of potential N₂O production in relation to total denitrification rates were more important in fertilized soils. Thus, fertilization reduces the relative importance of biotic controls of N₂O emissions, suggesting that biotic controls are weakened by elevated N levels." (L103-105)

Results and Discussion

Comment 9. L135: small r

A: fixed

Comment 10. L152: or they are outcompeted, but this would not necessarily mean they are related ecologically

A: We're uncertain about this comment but we wish to clarify that this statement was pertaining to lineages *within* each phylum, not that there was ecological coherence *across* the different phyla. We've made a revision (L163) to clarify this.

Comment 11. L166: but horizontal gene transfer was not tested here so don't over state.

A: We agree, although it is established that denitrification as a trait is incongruent with the organismal phylogeny. We have revised this statement to lessen the emphasis on HGT as also other evolutionary process are important (L181-182).

Methods

Comment 12. Likely some text can go in supplement, as is, is very long.

A: We agree and have now added a new Methods section in Supplemental Materials, and have shortened this section in the main text.

Comment 13. Figure: Would it be possible to add * next to clades that have known potential to have N related genes (from literature)

A: We appreciate the authors suggestion, however given the rather polyphyletic nature of denitrification and N₂O reducing genes, we fear this might be a bit misleading. There is a large body of literature (e.g. Jones et al., 2008; Graf et al., 2014) that demonstrate how even closely related strains may or may not have various gene associated with denitrification. Also see answer to comment on I.166 above.

Comment 14. Figure 4: ALE plots could be moved to supplement

A: It would be an option if it is important to save space. However, we think the ALE plots are important to show the direction of the relationships since this is crucial to understand the importance of various predictors. If transferred to SI, the reader would have to flip between the ms and SI to get the point.

Reviewer #2 (Remarks to the Author):

This paper used an impressive dataset to demonstrate legacy effects of N_r on N₂O emission via microbial structure. The authors collected 108 soil samples from 14 long-term fertilization field experiments and used qPCR & sequencing to investigate the genes relating to N₂O emission, and the microbial clustering driving N₂O emission via network analysis. They showed that N_r legacy altered the relative importance of biotic and abiotic predictors on potential N₂O emissions, and biotic factors are more important in unfertilized soils. The bioinformatic and statist analysis were well performed. I've a few comments below.

Comment 15. (1) N₂O from agricultural soils can also, as a byproduct, come from the ammonia oxidation process. In this paper, only the denitrification genes were investigated, while the amoA was not referred to. It might not be necessary to do extra work in the lab; but given the authors had sequencing data of 16S rRNA gene, is it possible to show a lit bit relating result via the ammonia oxidizers in Beta-proteobacteria group/Thaumarchaeota?

A: Yes, ammonia oxidation may also result in N₂O production and we appreciate the authors concern about not including information on ammonia oxidizing communities in this study. We did inspect the 16S rRNA gene data to determine abundances of both Thaumarchaeota and AOB, and after removal of rare OTUs we found 11 OTUs classified as Nitrososphaeraceae, and only single OTU of *Nitrosospira*. Abundances of AOA were not affected by fertilization, whereas the single, frequently occurring AOB OTU increased. We have mentioned this in the revised manuscript and highlight that it could add to in situ N₂O emission from fertilized soils (L174-178). However, we do not elaborate on this in the manuscript since the assay for determining N₂O production was conducted under anoxic conditions and therefore no N₂O produced could originate from ammonia oxidation. Thus, including the ammonia oxidizers in explaining the N₂O produced would be incorrect. Although ammonia oxidizers also produce N₂O, denitrifiers and nitrous oxide reducers are arguably the dominant communities controlling net N₂O emissions, even in soils below 60% water filled pore space (Azam et al. 2002, Biol Fertil Soils 35; Friedl et al. 2021, Environ. Res. Lett. 16; Liang and Robertson 2021, Glob Change Biol 27).

Comment 16. (2) The network analysis part is fantastic but also a bit confusing to me on two points. (a) Please unify the network name as co-occurrence network or co-association network. Given OTUs abundance were used, I think co-association is a more accurate term. (b) Fig3a: The colored segment, i.e., the taxonomic groups are not presented in every group, e.g., in common (-) network, only seven lines presents while most of the taxonomic groups do not present here. If possible, it might be better to remove those non-related color segments from each network. Just FYI.

A: As suggested, we have adjusted the text and use the term co-association consistently throughout the manuscript. However, we felt that the consistent use of colored segments for the various taxa across all networks highlights the dramatic differences we see amongst linkages that are common to both networks vs. those that are unique to unfertilized and fertilized networks. We would prefer to keep these as is in the figure.

Comment 17. Fig1: please describe all significant levels in the legend (the one of NirS.)

A: This has been added.

Reviewer #3 (Remarks to the Author):

Comment 18. The manuscript “Reactive nitrogen rewires and relaxes microbial controls of soil N₂O emissions” reported effects of Nr addition on soil N₂O production by analyzing soil denitrification potential, microbial community composition, and denitrification related functional gene abundance in 14 field trials with long term fertilization. The authors reported increased complexity of data association-based microbial networks under fertilization. They found that edaphic factors pH and Ca became important predictors of denitrification end product ratio in addition to biotic factors in fertilized soil. The central question discussed in this manuscript is interesting and important to the prediction of agricultural impact on climate change. The experiments and analysis performed are comprehensive in addressing the scientific question discussed. However, I found the wording at many places unclear and affects reading and understanding (see below detailed comments). In addition, since association network has an intrinsic problem in telling mechanisms underlying the links, results regarding the increased network complexity under fertilization, and biological meaning of separating communities within networks are descriptive. I hereby suggest revising the manuscript for clarity and accuracy.

A: We have tried to improve the text, see our answers to the specific comments below.

Detailed comments:

Comment 19. Title. I had a hard time interpreting “rewires microbial controls” before I read about the networks. I think the authors meant that the “links were rewired among nodes”. “Rewire” is not informative here in the title. From the results, I see abiotic factors pH and Ca became important predictors of denitrification end product ratio. However, there were still a few key microbial-related variables. So I think “relax” lacks accuracy.

A: We agree that it could be difficult to get the point in the title before reading the manuscript, especially regarding term rewires. We therefore propose a new title where we have replaced ‘rewire’ and ‘relaxes’ with ‘restructures’ and ‘weakens’: “Reactive nitrogen restructures and weakens microbial controls of soil N₂O emissions”. Weakens indicate that the biotic controls become less prominent. Also, rewire has been changed to restructure and relaxes to weakens throughout the manuscript.

Comment 20. L62-64. Explain “higher taxonomic groups”. I am confused about the reasoning behind this sentence. Are there reports of changes in abundances of organisms under fertilization that do not harbor N-related traits?

A: We have tried to clarify the reasoning here (L60-66) and also avoid using “higher taxonomic groups” (refers to higher taxonomic ranks). We wished to highlight that, according to the more recent work of Isobe et al. (2019), the use of a phylogeny-based approach to examine the response of microbial communities to elevated N is likely to be of greater utility in predicting N₂O emissions in response elevated N_r than assessing shifts in the abundances of taxonomic groups. Isobe et al. demonstrated that addition of N_r to grassland soils resulted in changes in the abundance of different lineages of bacteria that were not uniform within higher taxonomic ranks, e.g phyla or class level. This is in contrast to reports we refer to in L62 that describe the effect of elevated N on the abundances of major bacterial phyla or classes, which then concluded that organisms within the various taxonomic groups affected by elevated N have common life-history traits.

Comment 21. L117. abundance of the gene ...

A: Fixed

Comment 22. L190. not clear what “in a modular fashion” means

A: Modular means that organisms that perform denitrification often do not have the full repertoire of genes/enzymes required to convert the entire pathway, and thus the different steps of the denitrification pathway can be performed by different organisms. We have removed the term ‘modular’ and revised this section to clarify the concept (L204-207), which was also highlighted in the introduction (L48-50).

Comment 23. L192. How microbial association could influence N₂O emission? Please elaborate.

A: Since N₂O production and reduction steps can be split across different species within a microbial community, changes in microbial co-associations may influence N₂O emissions. With the modifications in the text mentioned in relation to the previous comment, this should now be clear.

Comment 24. L215. “More homogenized” does not mean “more connected”. Homogeneous networks refer to networks where all nodes have the same function (i.e., same connectivity). The authors did not report parameters related to network homogeneity.

A: We appreciate the reviewer pointing this out, and we have included two additional metrics in table 1, connectance and average Jaccard similarity across nodes, that should provide a means to quantify network homogeneity (Kaiser-Bunbury et al., 2015; Delmas et al., 2019). To our understanding, connectance describes the evenness in the number of links per node across nodes in the network, whereas the average Jaccard index describes the degree to which nodes in the network share similar partners. We found that both metrics increased in the fertilized network, indicating that OTUs within this network have a greater degree of ecological overlap with other OTUs. We have adjusted this text in both the results and methods sections in the manuscript (L234-240 and L492-493 respectively). For clarity, we have avoided the use of ‘homogenous’ and instead refer to OTUs in the fertilized network have a higher degree of ecological overlap, which we defined as arising from multiple possible mechanisms (L241-244).

Comment 25. L222. “increased interactions” refers to what?

A: We have clarified this as follows: “At the same time, the decreased path length [in the fertilized network] also supports a potential increase in even mutualistic interactions, such as cross-feeding of C or N substrates, as the inverse of this metric indicates a higher efficiency in the system” (L244-246).

Comment 26. L224-225. Generally, more available resource is expected to alleviate competition. Are some of the negative links connecting fast growers in the network for fertilized soils?

A: We agree with the reviewer’s statement that more available resource is expected to alleviate competition, particularly if different species are limited by different resources. However, we think this relates more to resource complexity, as per the Resource ratio theory. Competition may also occur when there is an overall increase in resource *concentration/content* (Ghoul and Mitri, 2016 Trends in Microbiology). While we cannot speculate on whether resource complexity has been altered by long-term N addition, the overall increase in organic C, NO₃ and NH₄⁺ in the fertilized soils would indicate that, in general, the resources contents are indeed higher in the fertilized soils, and it is not unreasonable to suspect that such conditions would likely favor fast growers (i.e. copiotrophs) than slow grower (oligotrophs). For example, we found that the percentage of total negative links that were only between oligotrophic and copiotrophic taxa was 22 % in the unfertilized network, and increased to 28 % in the fertilized network. This is based on classifying different taxonomic groups using the consensus of the literature outlined (Ho et al. 2017), in which classes of Actinobacteria, Alpha-, Beta- and Gammaproteobacteria, and Bacteroidetes were classified as copiotrophic taxa, while Acidobacteria, Deltaproteobacteria and Verrucomicrobia were oligotrophic.

Comment 27. L228. Again, “Homogenizing effect” is confusing here.

A: We have also removed the term ‘homogenizing effect’ here to avoid confusion and revised the text (L250-252).

Comment 28. L229. How does increased complexity point to a more dynamic community?

A: We have deleted this since the explanation would be too speculative

Comment 29. L232. Please explain “network alignment”.

A: We thank the reviewer for highlighting this statement, as we realize that the use of the term ‘alignment’ was not accurate. We have therefore revised this statement to more accurately describe the use of the DyNet software as a means of network comparison, allowing us to identify nodes and linkages that are shared between the two networks (L254-256).

Comment 30. L236-247. This paragraph talked about modules in the networks, but what are the implications? From Figure S4, the common modules A, B, C and D contained many unique nodes, and the unique modules also contained many common nodes. What is the criterion used to define if a module is shared by two networks or not? This poses a problem in the interpretation of results in Figure 3c and Figure 4, especially for the common modules in the two networks since these modules are very different in the two networks.

A: We thank the reviewer for helping us clarify this aspect. Detection of network modules was done independently for each network using the edge-betweenness algorithm, as described in the materials and methods (L501-502). After which, the DyNet software package was used as described in the text (L254-256; L501-502) to examine the degree of overlap between the modules identified separately in each network. We then identified shared modules between the two networks graphically based on the merged network structure presented in Fig. S4. We observed that OTUs that were either lost or gained from the large modules due to fertilization were *not* members of other detected modules, with the exception of the five nodes that overlap in modules A and B, and those in B and fC. This, plus the fact that these nodes are connected only to the ‘core’ of common nodes in each module, we felt it appropriate to consider these as shared yet restructured modules between the two networks, rather than completely different modules. We have adjusted the text in L262 to reflect this. In contrast, common nodes five of the unique modules were ‘shuffled’ between different detected communities in each network. We therefore consider these to be unique modules, and show the degree of overlap in figure S4 and discuss this in L268-272.

Comment 31. L267-268. But two of them showed no correlation with $N2O/[N2+N2O]$ ratio in Figure 3c.

A: We thank the reviewer for catching the error, and have corrected this statement in the text (L291-292).

Comment 32. L317. Define ALE.

A: This is defined in the methods (L524-527) section and have now also been added in the Results and Discussion to help the reader (L342-343).

Comment 33. L793. composition of communities....

A: This has been corrected.

Comment 34. Figure 2. Add legends to node symbol size and branch color hue. Also, it is hard to identify changed clades. Circles are at the fork and not clear which clade or both clades after the fork changed.

A: A legend has been added to this figure. To clarify, the PhILR method transforms OTU abundance data to phylogenetic balances to deal with data compositionality. This means that, when a significant effect of fertilization is observed, the balance in abundance has shifted between sister clades. Thus, one clade decreases while the other increases. We refer the reviewer to Silverman et al., 2016, Washburn et al. 2017 and Knight et al., 2018 for additional information on this method.

Fig. 2

Comment 35. Figure 3. Add a legend for strengths of correlation

A: This has been added as indicated.

Comment 36. Table S1. Fertilization amounts unclear

A: This has been clarified in the table.

Comment 37. Figure S4. I don't see grey edges. What are the percentages of the common/unique nodes and links?

A: To help clarify the reviewer comment above about shared vs. unique modules, we have added this information into the figure in the table as indicated below, and adjusted the grey edges in the network to make them more visible.

Module	Common (%)		Unfertilized (%)		Fertilized (%)	
	Nodes	Edges	Nodes	Edges	Nodes	Edges
A	61.7	29.2	4.2	10.5	34.1	60.3
B	53.8	16.0	12.1	20.7	34.1	63.3
C	55.6	26.6	34.6	40.3	9.9	33.1
D	75.0	22.9	18.1	59.6	6.9	17.5
uA	57.8	15.4	42.2	84.6	na	na
uB	75.0	50.8	25.0	45.2	na	na
uC	38.9	0.0	61.1	100.0	na	na
uD	15.4	4.5	84.6	95.5	na	na
uE	33.3	0.0	66.7	100.0	na	na
uF	33.3	20.0	66.7	80.0	na	na
fA	100.0	55.6	na	na	10.0	44.4
fB	50.0	20.0	na	na	50.0	50.0
fC	51.4	23.2	na	na	48.6	76.6
fD	33.3	12.5	na	na	66.7	87.5
fE	85.7	25.0	na	na	14.3	75.0

REVIEWERS' COMMENTS:

Reviewer #2 (Remarks to the Author):

The authors had addressed my previous concerns, and from my side, I recommend the paper considered for publication by Communications Biology.

Reviewer #3 (Remarks to the Author):

The authors have fully addressed my concerns and questions raised based on the previous version of the paper. The conceptual figure is a nice summary of the paper.

Typo: L231. were higher than